# TriCCo v1.1.0 - a cubulation-based method for computing connected components on triangular grids

Aiko Voigt[1], Petra Schwer[2], Noam von Rotberg[2], and Nicole Knopf[3]

[1]Department of Meteorology and Geophysics, University of Vienna, Austria
[2]Institute for Algebra and Geometry, Department of Mathematics, Otto-von-Guericke University, Magdeburg, Germany
[3]Institute of Meteorology and Climate Research - Department Troposphere Research Karlsruhe Institute of Technology, Germany

**Correspondence:** Aiko Voigt (aiko.voigt@univie.ac.at)

**Abstract.** We present a new method to identify connected components on triangular grids used in atmosphere and climate models to discretize the horizontal dimension. In contrast to structured latitude-longitude grids, triangular grids are unstructured and the neighbors of a grid cell do not simply follow from the grid cell index. This complicates the identification of connected components compared to structured grids. Here, we show that this complication can be addressed by involving the mathematical tool of cubulation, which allows one to map the 2-d cells of the triangular grid onto the vertices of the 3-d cells of a cubic grid. Because the latter is structured, connected components can be readily identified by previously developed software packages for cubic grids. Computing the cubulation can be expensive, but importantly needs to be done only once for a given grid. We implement our method in a Python package that we name TriCCo and make available via pypi, gitlab and zenodo. We document the package and demonstrate its application using simulation output from the ICON atmosphere model. Finally, we characterize its computational performance and compare it to graph-based identifications of connected components using breadth-first search. The latter shows that TriCCo is ready for triangular grids with up to 500,000 cells, but that its speed and memory requirement should be improved for the application to larger grids.

## 1 Introduction

Climate and atmospheric modeling is experiencing a leap in its ability to represent Earth digitally (Satoh et al., 2019; Wedi et al., 2020). The leap is made possible by a drastic increase in spatial resolution and the development of global storm-resolving models that apply local differencing schemes and discretize the sphere by means of unstructured grids. An example of the latter is the triangular grid based on the icosahedron and applied in the ICON unified weather and climate model (Zängl et al., 2015; Giorgetta et al., 2018). The triangular grid is a defining difference of ICON to its predecessor models ECHAM and COSMO, which were based on latitude-longitude grids.

While having many numerical advantages, the change from a structured latitude-longitude to an unstructured triangular grid challenges established workflows and analysis methods. For some types of analysis one might accept

to interpolate the model output to latitude-longitude coordinates. For others, however, an interpolation might be problematic as it artificially smoothes the boundary of objects, thereby potentially introducing an ambiguity in object-based analyses.

One analysis that is ideally done on the native model grid is connected component labeling. In atmospheric sciences, connected component labeling is applied for object-based studies of atmospheric moisture, clouds and their topology. For example, previous work used it to characterize large-scale moisture transport in the form of atmospheric rivers (Muszynski et al., 2019) and to study clusters of convective clouds (Neggers et al., 2003; Rieck et al., 2014; Rempel et al., 2017; Licón-Saláiz et al., 2020), whose size statistics and distance to neighbors impacts cloud behavior, cloud organization and cloud radiative effects (Schäfer et al., 2016; Jakub and Mayer, 2017). With atmosphere and climate models moving to storm-resolving resolutions of a few kilometers and finer, three-dimensional radiative effects of clouds are becoming increasingly important. Because the radiative properties of clouds differ strongly from those of their surrounding air, a connected-component labeling that respects the sharp boundaries between cloudy and cloud-free air seems especially important.

When connected component labeling is done on structured latitude-longitude grids, one has access to widely used analysis tools such as opencv (Bradski, 2000) and scikit-image (van der Walt et al., 2014) that are well documented and easily found. This comfort is lost when working on triangular grids. In fact, this loss has sparked our collaboration that addresses a need from climate modeling (A.V.) and draws on expertise from pure mathematics (P.S.). Before we started our collaboration, one of us (A.V.) had – without much success – sought advice from some colleagues in climate science and visualization science regarding a tool for connected component labeling on a triangular grid. When revising the paper and thanks to the input of the reviewers it has become clear to us that graph-based solutions provide such a tool (see Sect. 5). But the fact that this was not clear neither to us nor our colleagues in our view is an anecdotal illustration of how unstructured triangular grids challenge traditional workflows in climate science that have been used for many decades and now need to be revised in the light of storm-resolving models.

In this paper we present a new method that lifts a triangular grid to a cubic grid by means of cubulation. The method takes data that is stored as an unordered 1-dimensional array and indexed in terms of triangles, and makes this data accessible in the form of a three-dimensional matrix, i.e., a cubic grid. As a result, neighbor relations between triangles are encoded in the indices of the cubical grid and become self evident, and analysis tools developed for 3-dimensional image analysis (e.g., in computer vision or neuroimaging) can be employed to the data's three-dimensional matrix representation. This is the key idea and motivation of our method. It is also worth noting that the cubic structure gives rise to fast and efficient algorithms to compute, for example, shortest paths, which has been applied for moving robotic arms or studying the space of potential gene or language mutations (Ardila et al., 2014).

Our method targets a specific analyis problem, yet in the larger context of climate and atmospheric modeling we also hope that our work helps address the need for new strategies in analyzing the big-data output from the next generation of climate models. Fig. 1 illustrates this need by comparing the simulation of clouds over the North Atlantic for two model resolutions (Senf et al., 2020). The left panel shows the cloud field for the model run at a 80 km

resolution, for which the triangular grid structure can be spotted by eye. The resolution is typical for contemporary global climate models used to anticipate how climate will evolve over the coming century (Eyring et al., 2016). The right panel shows the same cloud field simulated at a much finer resolution of 2.5 km and illustrates the rich patterns of clouds at the mesoscale that are becoming accessible in storm-resolving models.

The purpose of the paper is threefold. In Sect. 2 we first introduce the basic idea of our method and rigorously define its mathematical foundation. Secondly we practically implemented the method and developed an open-source Python package named TriCCo. The implementation is described in Sect. 3, and an example for an application is presented. The third purpose is to characterize the strengths and weaknesses of TriCCo's current implementation in Sect. 4, so as to both demonstrate its feasibility as well as to point out how its computational performance can be improved. To this end, we also compare TriCCo to alternative approaches based on breadth-first search and graphs in Sect. 5. We discuss possible extensions of TriCCo and conclude in Sect. 6. A reader mostly interested in the use of TriCCo might focus on Subsect. 2.1 and Sects. 3 and 4.

## 2 Methodology: Component labeling via cubulation

### 2.1 General idea

Before detailing the mathematical aspects of our method, we describe its general idea in this subsection. The method is based on the realization that a triangular grid can be embedded into a 3-dimensional cubical grid. On the triangular grid, the cell centers are indexed as a 1-dimensional array that on its own does not describe the neighbor relationships. On the cubical grid, in contrast, the triangles are indexed in terms of triples $(x, y, z)$, and the neighbor relations become self evident.

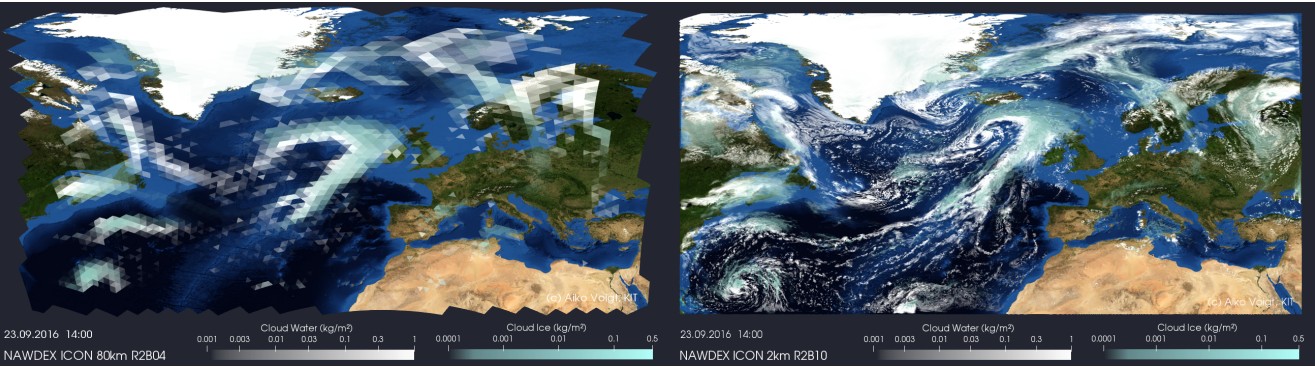

**Figure 1.** Illustration of clouds simulated by the ICON atmosphere model in a low-resolution version with 80 km horizontal grid spacing (left) and a high-resolution version with 2 km horizontal grid spacing (right). The triangular grid structure is visible in the left panel.

The simultaneous, adjacency preserving translation of cell indices on the triangular grid into $(x, y, z)$-positions on the cubical grid is called cubulation. This method makes use of the three sets of parallel classes of lines (also called hyperplanes) in the grid that are formed by the edges of the triangles. Each set of parallel lines is consecutively numbered and the position of a triangle can be described by the three indices of the lines that contain the triangle's edges. This process leads to the cubical coordinates.

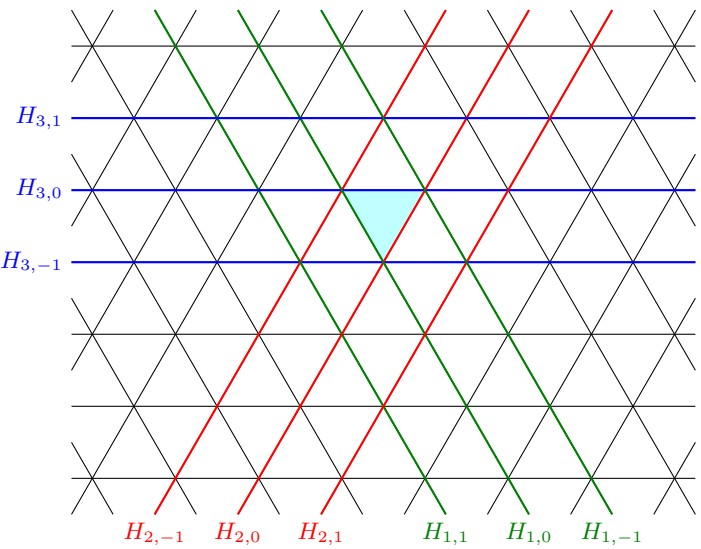

**Figure 2.** Illustration of the general idea of the cubulation method. The triangular grid is shown in terms of the triangle edges in black. Some parallel hyperplanes are shown in the same colors.

The concept is illustrated in Fig. 2. The edges of the triangle cells are shown in black, and the base triangle is highlighted in cyan. Each edge of the base triangle is contained in a unique line in the plane. The three lines obtained in this way are highlighted in red, green and blue. Each other line is parallel to one of the three lines. We enumerate these classes from one (green) to three (blue). Some examples of parallel lines are highlighted in the respective color. We index each line by a number as shown in Fig. 2 by starting with the index 0 for the line containing the edge of the cyan triangle. The position of any triangle in the grid is described by means of the line indices of the hyperplanes. For example, the position of the highlighted triangle is $(0, 0, 0)$. The three neighbors that share an edge with the highlighted triangle have indices $(1, 0, 0)$ (lower-left triangle), $(0, 1, 0)$ (lower-right triangle) and $(0, 0, 1)$ (upper triangle). For a precise description, see Definition 2.6.

As a result, the neighbors of a triangle are self-evident when the cubical positions are used, and the connected component labeling can be performed on a structured cubical grid.

## 2.2 Mathematics of the cubulation method

In this subsection we describe the algorithm that transforms (a connected subset of) the regular triangle tiling of the Euclidean plane into a subset of the standard subdivision of $\mathbb{R}^3$ into unit cubes. The method is a concrete example and implementation of Sageev's cubulation method introduced in Sageev (1995) for the Coxeter group of type $\tilde{A}_2$. The vertices of this cubulation will have integer-valued coordinates. We start with some characteristics concerning the structure of the triangular grid in Section 2.2.1, collect some necessary background on cube complexes in Section 2.2.2 and then carry out the construction in Section 2.2.3

### 2.2.1 Structure of the triangle tiling

The cubulation of the regular triangle tiling of the plane is the key tool that makes TriCCo work. The regular triangle tiling of the plane will be called $\Sigma$ in the following. The space $\Sigma$ carries the structure of a (metrized) simplicial complex whose maximal simplices are all 2-dimensional and in which all edges have the same length. A picture of this complex is provided in Figure 3 below. The 2-dimensional simplices are the triangles in this figure; edges are shown in black.

The cubulation that we construct restricts to any (connected) subset of the triangles of the plane and hence automatically yields a method to cubulate local lattices of any granularity.

There are three parallel classes of lines in $\Sigma$. We have illustrated these classes in Figure 2 by highlighting some lines of the same class in the same color. These three classes will correspond to the three pairwise perpendicular coordinate axes of $\mathbb{R}^3$ in which the cubulation lives.

We will now define a graph associated with $\Sigma$.

**Definition 2.1.** The *dual graph* $\Gamma_\Sigma$ of the triangle tiling $\Sigma$ of the plane is defined as follows: The set of vertices $V$ in $\Gamma_\Sigma$ is the set of triangles in $\Sigma$. There exists an edge $(u,v)$ between $u,v \in V$ if and only if the triangles $u$ and $v$ share a codimension one face, i.e., they have an edge in common.

The dual graph can be pictured inside the tiled plane as follows. Draw a point in the center of each of the triangles. Each of these points represents a vertex of the graph $\Gamma_\Sigma$. Two points are connected by an edge whenever the corresponding triangles have a side in common. These edges may be drawn perpendicular to the common face. Figure 3 illustrates this correspondence.

Each vertex of the dual graph $\Gamma_\Sigma$ by construction corresponds to a unique triangle in $\Sigma$. Every hexagon in $\Gamma_\Sigma$ corresponds to a collection of six triangles in $\Sigma$ sharing a common vertex.

### 2.2.2 Cubical complexes

Cubical complexes are spaces obtained by gluing unit cubes of various dimensions along isometric faces, i.e. faces of the same dimension. A unit cube is a cube in some Euclidean space of dimension $k$ all of whose edges are of length

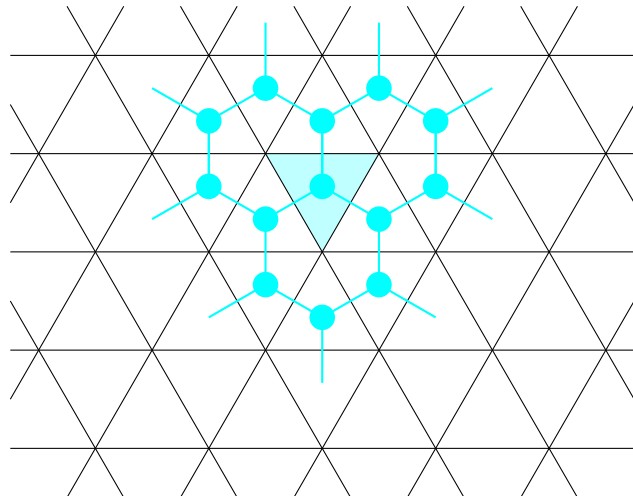

**Figure 3.** The figure shows a piece of the equilateral triangle tiling $\Sigma$ of the plane. The turquoise vertices and edges represent the dual graph of the tiling as defined in Definition 2.1.

one. More formally and in short, a *cubical complex K* is an $M_k$-polyhedral complex such that all the shapes are unit cubes, i.e. of the form $[0,1]^k$ for some $k \in \mathbb{N}$. For details on $M_k$ complexes see Bridson and Haefliger (1999) or Schwer (2019).

We will provide an ad-hoc definition of a cubical complex below in order to allow for a treatment of the subject without the need of introducing general $M_k$-polyhedral complexes.

**Definition 2.2** (Cubes). An *n-cube* is a set $C$ of the form $C = [0,1]^n \subset \mathbb{R}^n$. A *codimension one face* of $C$ is given by $F_{i,\epsilon} := \{x \in C | x_i = \epsilon\}$ for $\epsilon \in \{0,1\}, i = 1, \dots, n$. All other (proper) faces of $C$ are non-empty intersections of codimension 1 faces. We say that $x \in C$ is an *inner point* of $C$ if $x$ is not contained in any (proper) face of $C$.

These cubes will now be glued together to form larger complexes. For technical reasons we will assume that the intersection of two cells in a cubical complex is either empty or a face of both. Some, but not all, of the gluings that do not satisfy this assumption can be resolved by further subdividing the complex into smaller cubes.

**Definition 2.3** (Cubical complexes). Let $C$ and $C'$ be two cubes with faces $F \subseteq C$ and $F' \subseteq C'$ [1]. A *gluing* of $C$ and $C'$ is an isometry $\phi : F \to F'$, which provides an identification of two of the sides of the cubes.

Suppose $\mathcal{C}$ is a set of cubes and $\mathcal{S}$ a family of glueings of elements of $\mathcal{C}$, that is for all $C \in \mathcal{C}$ there is $n_C \in \mathbb{N}$ such that $C \cong [0,1]^{n_C}$ and every $\phi \in \mathcal{S}$ is an isometry $\phi : F \to F'$ where $F, F'$ are faces of cubes $C, C' \in \mathcal{C}$. Assume further that $(\mathcal{C}, \mathcal{S})$ satisfies the following two conditions:

1. No cube is glued to itself.

---
[1]Note that here possibly $F = C$ or $F' = C'$

2. For all $C, C' \in \mathcal{C}$ there is at most one gluing of $C$ and $C'$.

Then $(\mathcal{C}, \mathcal{S})$ defines a *cubical complex* $(X, d)$ by putting $X := (\bigsqcup_{C \in \mathcal{C}} C) \big/ \sim$ where $\sim$ is the equivalence relation generated by putting $x \sim \phi(x)$ for $\phi \in \mathcal{S}$ and $x \in \mathrm{dom}(\phi)$. The metric $d$ on $X$ is the length metric induced by the restricted Euclidean metric on each cube in $\mathcal{C}$.

An example of a cubical complex is shown in Fig. 4.

One property of a cubical complex $X$ is that the restriction of the quotient map $\mathrm{p} : \bigsqcup_{C \in \mathcal{C}} C \to X$ to one cube $C \in \mathcal{C}$ is injective. And that the intersection of two cubes in $X$ is either empty or a face of both (here a face might be the whole cube). Hence we may identify a cube $C \in \mathcal{C}$ with its image in $C$ and write $C \in X$.

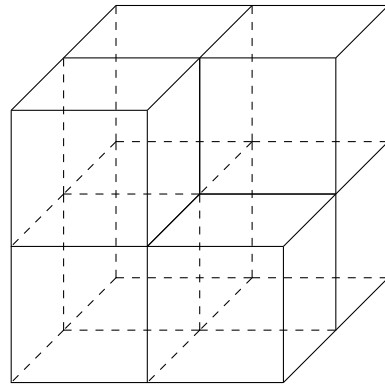

**Figure 4.** A cubical complex built out of seven 3-cubes.

One of the key features of a cube complex are *hyperplanes*. Hyperplanes are cubical complexes themselves which
we may associate to the midcubes parallel to codimension-one faces of certain cubes. These hyperplanes then cut through the middle of adjacent cubes. Examples are shown in Figure 5.

In a cube complex that satisfies the additional curvature property of being CAT(0) every hyperplane cuts the complex into two disjoint pieces, called halfspaces. The partially ordered set of all these halfspaces allows to recover the cubical complex itself.

In the next section we will cubulate the equilateral triangle tiling of the plane using the hyperplanes and half-spaces appearing in the tiling. More generally one can introduce an abstract notion of half-space systems and use those to cubulate more abstract spaces than the example we are considering here. See for example Schwer (2019).

Our main goal is the following.

**Main Goal 2.4.** *Construct from every edge-connected subcomplex $A$ of $\Sigma$ a subcomplex $X(A)$ of the standard*
*cubulation $X$ of $\mathbb{R}^3$. Adjacency of triangles in the plane should be equivalent to adjacency of the associated cubes in $X$.*

As mentioned above, Euclidean 3-space can be subdivided and equipped with the structure of a (metric) cubical complex. We call this cube complex the *standard cubulation* of $\mathbb{R}^3$ and denote it by $X$. Its vertices are the points in

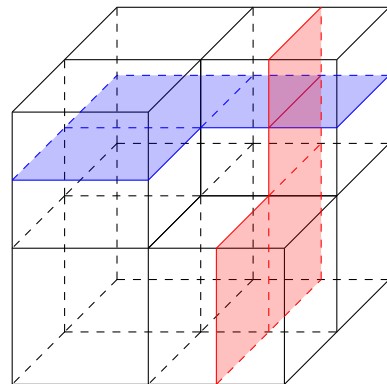

**Figure 5.** The blue and red 2-dimensional cubical complexes are examples of hyperplanes in the cubical complex we have already seen in Fig. 4.

$\mathbb{R}^3$ whose coordinates are all integers with respect to the standard basis $\{(1,0,0),(0,1,0),(0,0,1)\}$. Denote this set of vertices by $X^{(0)}$. The edges of the cubes are the intervals between any pair of these integer valued vertices that differs in exactly one entry. The graph that is formed by these vertices and edges is called the *one-skeleton* of $X$ and is denoted by $X^{(1)}$.

### 2.2.3 Construction of the cubulation of $\Sigma$

In fact we will not cubulate $\Sigma$ but its dual graph $\Gamma_\Sigma$. To be precise: The goal is to define a simplicial map from $\Gamma_\Sigma$ to the one-skeleton $X^{(1)}$ of the standard cubulation $X$ of $\mathbb{R}^3$.

We now introduce a labeling of the lines in the tiling $\Sigma$ which will allow us to define such a map.

**Definition 2.5.** A *consistent labeling* of the set of hyperplanes in $\Sigma$ is the following procedure that assigns to every line the number of its class and an integer index as follows. Fix a base-triangle $v_0$ in $\Sigma$. There exist then three hyperplanes each containing one of the three sides of $v_0$. Call them $H_{1,0}, H_{2,0}$ and $H_{3,0}$, respectively. In addition there is a unique hyperplane parallel to $H_{i,0}$ whose intersection with $v_0$ is a single vertex. Call this hyperplane $H_{i,-1}$ and enumerate all other hyperplanes in the same parallel class periodically.

See Figure 2 for an illustration of the labeling we have just defined. In the next definition we obtain from the labeling defined in Definition 2.5 coordinates for the vertices in $\Gamma_\Sigma$. These can be used these to define a map from the vertex set $V$ of $\Gamma_\Sigma$ to the vertices $X^{(0)} \subset X$.

**Definition 2.6** (The 3d-coordinates for triangles)**.** Recall that $V$ is the set of vertices of the dual graph $\Gamma_\Sigma$. For each $v \in V$ define 3-dimensional *coordinates* $(v_1, v_2, v_3)$ by putting $v_i := k$ if the triangle in $\Sigma$ corresponding to $v$ lies between the hyperplanes $H_{i,k}$ and $H_{i,k-1}$ in $\Sigma$.

In Figure 3 the dual graph $\Gamma_\Sigma$ is shown in turquoise. The vertex of the dual graph inside the turquoise triangle has coordinates $(0,0,0)$. Vertices contained in a common hexagon of the dual graph will be mapped to the same 3-cube in the cubulation.

**Definition 2.7** (Cubulation map)**.** The *cubulation map* $f : V \to X^{(0)}$ is defined by $v \mapsto (v_1, v_2, v_3)$ where the $v_i$ are chosen as in Definition 2.6.

Figure 6 illustrates some of the images of vertices in $\Gamma_\Sigma$ inside the 1-skeleton of $X$.

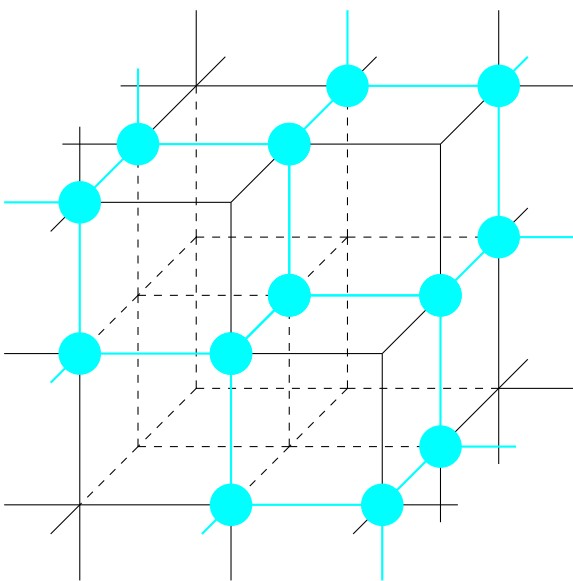

**Figure 6.** Illustration of the mapping from $\Gamma_\Sigma$ to $X^{(1)}$ showing how the dual graph sits inside the 3-dimensional cubical complex.

The cubulation map satisfies some properties and in particular preserves adjacency of vertices.

**Proposition 2.8** (Properties of the cubulation map)**.** *Let $f$ be the map defined in Definition 2.7. Then the following holds.*

1. *The map $f$ preserves adjacency, that is, the coordinates of two adjacent vertices $u,v$ in $\Gamma_\Sigma$ differ in exactly one entry. Their images in $X^{(1)}$ under $f$ are connected by an edge.*

2. *Every hexagon in $\Gamma_\Sigma$ is mapped into a unique cube of $X$.*

3. *Triangles $u,v$ that share a vertex in $\Sigma$ are mapped to vertices that are contained in a same cube.*

*Proof.* To see the first item let $u$, $v$ be adjacent vertices in $\Gamma_\Sigma$. They correspond then to two triangles that share a common side. This side is contained in a unique hyperplane $H_{i,k}$ for some parallel class $i$ and some index $k$. So the

coordinates $v_i$ and $u_i$ differ by one. If there was a second coordinate in which $u$ and $v$ would differ, there would be a second hyperplane in a different parallel class separating $u$ and $v$. But this is impossible.

By checking one of the hexagons by hand one can verify that the second property is satisfied and all vertices of this hexagon are mapped to a common cube. The vertices in all other hexagons have hyperplane coordinates shifted by integer values in at least one of the three directions obtained from the parallel classes of hyperplanes. This yields the assertion.

The third item follows from the second by checking that triangles sharing a vertex are contained in a common hexagon in $\Gamma_\Sigma$. $\qquad\square$

We can characterize the full image of $f$ and describe which points in $X^{(1)}$ are part of the embedded graph $\Gamma_\Sigma$.

**Proposition 2.9.** *The image $f(V)$ of all the vertices in $\Gamma_\Sigma$ is the collection of points in $X^{(1)}$ whose coordinates sum up to either $1$ or $0$. Each edge $(u,v)$ has a vertex with coordinate sum $0$ and one with coordinate sum $1$.*

*Proof.* Let $u$ and $v$ be two triangles sharing an edge. For each $i$ there is an index $k_i$ such that $u$ lies between $H_{i,k_i}$ and $H_{i,k_i-1}$ and $u$ has coordinates $(k_1, k_2, k_3)$. Observe that $v$ lies between the same two parallel hyperplanes for two of the indices. Moreover, there is one index, say $j$, for which $v$ is either between $H_{j,k_j+1}$ and $H_{j,k_j}$ or between the two hyperplanes $H_{j,k_j-1}$ and $H_{j,k_j-2}$.

Without loss of generality, let $j = 1$. Suppose first that $v$ is between $H_{1,k_1}$ and $H_{1,k_1+1}$. Then $v$ has coordinates $(k_1 + 1, k_2, k_3)$ and the coordinate sum differs by one. In case $v$ is between $H_{1,k_1-1}$ and $H_{1,k_1-2}$ it has coordinates $(k_1 - 1, k_2, k_3)$ and the coordinate sum differs by $-1$.

It remains to prove that coordinate sums alternate between $0$ and $1$. The base triangle $v_0$ has coordinates $(0, 0, 0)$ and hence coordinate sum $0$. Its neighbors lie, by construction, between hyperplanes with indices $0$ and $1$ in one of the three directions and have hence coordinate sum $1$.

One can proceed by induction on the distance to $v_0$ in $\Gamma_\Sigma$ and prove that along any shortest path in $\Gamma_\Sigma$ connecting an arbitrary vertex to $v_0$ the coordinates sums alternate between $0$ and $1$. $\qquad\square$

## 2.3 Identifying connected components for 2-d data

After having computed the cubulation of the triangular grid we can use it to detect connected components.

There are two ways to define connectivity on a triangular grid: either via shared edges (edge-connectivity) or via shared vertices (vertex-connectivity). The two types of connectivity can also be defined more rigorously as follows. A subset of the triangles of the triangle tiling $\Sigma$ is *edge-connected* if for any two triangles the corresponding vertices in the dual graph are connected by a path in the dual graph. We say that a set $A$ of triangles is *vertex-connected* if for every pair of triangles $u$ and $v$ in $A$ there exists a sequence of triangles $v_i \in A, i = 0, \ldots, n$ connecting $u$ and $v$ such that two subsequent triangles $v_i$ and $v_{i+1}$ share a vertex.

We need to to clarify how connectivity for cells on the triangular grid translates to connectivity for the cubical grid. One can show that edge connectivity on the triangular grid corresponds to face-connectivity (also known

as 6-connectivity) on the cubic grid and that vertex-connectivity on the triangular grid translates precisely to vertex-connectivity (also known as 26-connectivity) on the cubic grid.

## 2.4 Identifying connected components of 3-d data

We now describe how connectivity is computed for 3-dimensional data. An example of 3-dimensional data is cloud fraction. Cloud fraction depends on the horizontal (i.e., geographical position), which is described by the triangular grid, and altitude. To represent the vertical dimension, the ICON model stacks layers of horizontal grids. This treatment of the vertical dimension is standard in atmospheric modeling.

We understand connectivity in the vertical dimension to mean that neighboring layers share at least one triangle on the horizontal grid. E.g., if a cell $i$ corresponding to latitude-longitude position $(\varphi, \lambda)$ has the same value at model level $k$ and model level below, $k+1$, then the grid volumes spanned by $(k, i)$ and $(k+1, i)$ are connected. We note that we thus limit connectivity in the vertical to cell faces. This is consistent with the treatment of vertical exchange typical in atmospheric models, which occurs column-wise apart from a few exceptions such as 3-dimensional radiative transfer.

With this, connected components in 3-d can be computed from a three-step procedure. In a first step, 2-d components are identified for each model level separately following the method described in Subsect. 2.3. Each 2-d component is considered a node of an undirected graph. In a second step, we identify all pairs of 2-d components that reside in neighboring model levels and share at least one triangle. The pairs are edges between the nodes formed by the 2-d components. In a third step, we apply a connected component analysis on these nodes and edges formed by the set of 2-d connected components. The overall result of this procedure is a list of 3-d connected components, where each 3-d connected component is given by a list of 2-d connected components. For the connected component analysis of step three we use the external library NetworkX implemented in Python (Hagberg et al., 2008), but our procedure would also work for other external network analysis libraries.

## 3 Software implementation and application examples

In this section we describe the software implementation and provide basic examples on how to apply the method to cloud fields simulated by the ICON atmosphere model. Our aim is to provide an orientation on the code structure and its usage. Version 1.1.0 of the implementation described and used here is available via gitlab and pypi, and long-term archived at zenodo (see code availablility; Voigt, 2022).

The implementation of TriCCo and its use consist of four steps. Each step is described in the following. Note that the terms triangle and cells are used interchangeably, with no risk of confusion as our method is designed for triangular cells.

## 3.1  Step 1: Preparing the horizontal grid

Regarding the horizontal triangular grid, information on the neighboring cells and the edges of each cell is needed, as well as information on the vertices that form a specific edge. The grid information is stored in an xarray dataset (Hoyer and Hamman, 2017) named `grid` using variable names that follow the convention of the ICON model grid. The variable naming is due to the fact that the ICON grid was used during the development and testing of code. The code includes routines specific to the ICON grid. It should be straightforward to adapt the routine to other grids.

Let us assume that the grid consists of $n_c$ cells, $n_v$ vertices and $n_e$ edges. For a triangular grid that covers the entire sphere, $n_v$ and $n_e$ are given by $n_c$ as described in Zängl et al. (2015); for a limited-area grid, the relationships hold in an approximate manner. The three variables required to describe the grid are:

- `neighbor_cell_index` defines the three neighboring cells for each cell. The dimension is $(3, n_c)$.

- `edge_of_cell` defines the three edges for each cell. The dimension is $(3, n_c)$.

- `edge_vertices` defines the two vertices for each edge. The dimension is $(2, n_e)$.

The variables are indexed starting from 0. In ICON this requires a shift by $-1$ as the indexing starts with 1. The variables are accessed by three analogously-named functions that provide the variable values for a single grid cell or edge. For a limited-area grid, a missing neighboring cell indicates the grid boundary and is assigned a value of $-9999$.

## 3.2  Step 2: Computing the cubulation of the horizontal grid

The main function is `compute_cubulation`. This function implements the cubulation described in Section 2 and computes for each grid cell $i$ the associated 3-d coordinate on the cubic grid $(x, y, z)$. This information defines the cubulation and is stored in `cube_coordinates` as list of arrays of the form $(i; (x, y, z))$.

The function `compute_cubulation` starts at a user-specified `start_triangle`, and iteratively computes all cube coordinates within an expanding circle around the start cell. The iteration stops when the circle reaches a user-specified `radius`. The radius needs to be chosen according to the grid size, or alternatively can be set to a smaller value if only a specific part of the grid is of interest. If the radius is too small, the cubulation will not cover the entire grid. On the other hand, if the radius is too large, the algorithm will iterate over empty lists for the last steps. Setting `print_progress=True` outputs the progress of the iteration to screen, allowing one to monitor the number of new cells added in each round, which is helpful for choosing the radius. Also, even though iterating over empty lists comes with essentially no computational burden, the size of the cubulation increases with the radius, which in turn increases the memory demand of the connectivity analysis in Step 4. The radius thus should be as small as possible.

The following consideration is helpful when choosing the radius $r$. Each iteration adds $3 \cdot i$ cells, where $i \leq r$ is the number of the current iteration. The total number of visited cells is

$$n_c = 1 + \sum_{i=1}^{r} 3 \cdot i = 1 + 3 \cdot \frac{r(r+1)}{2} = 1 + 1.5r + 1.5r^2. \tag{1}$$

The sum begins with 1 due to the start cell. Thus, covering $n_c$ cells requires a search radius of

$$r = -\frac{1}{2} + \sqrt{\frac{1}{4} + \frac{2}{3}(n_c - 1)} \tag{2}$$

The equation is exact as long as the iteration has not reached the grid borders, i.e., it works best for circle-shaped grids such as those used by Schemann and Ebell (2020). For other grids, the equation serves as a lower bound of the
295 radius that one needs to cover $n_c$ cells. Acknowledging that $n_c \gg 1$, the lower bound can effectively be approximated by $r \gtrsim \sqrt{\frac{2}{3}n_c}$.

Another helpful approach to find the search radius is to start from a value somewhat larger than the lower bound and adapt the radius based on the diagnostic output of `compute_cubulation` that can be obtained by `print_progress=True`.

A few aspects of `compute_cubulation` warrant further description. The function begins by assigning the cube coordinate $(x,y,z) = (0,0,0)$ to `start_triangle`, but this is not the final coordinate of the start cell as explained further below. In each iteration, all 'new' cells that are adjacent to already visited cells are considered and their cube coordinates are calculated. Missing neighbors, as they occur for cells at the border of the grid, are identified by -9999 and ignored (cf. Section 3.1).

Moreover, the edges of a new cell need to be colored, and this needs to be done such that the edge colors are consistent with the edge colors of other cells. I.e., parallel edges need to have the same color as they belong to the same hyperplane. This is illustrated for two neighboring cells in Figure 7, where the left cell is `old` and the right cell is a so far unvisited neighbor `new`. The joint edge is already colored as `old` was visited in the preceding iteration. This leaves the task of coloring the two non-joint edges of `new`. If only one edge is uncolored, its color is given by the
color which has not yet been used. If two edges of `new` need to be colored, their colors are deduced from the edge colors of `old`: a non-joint edge of `new` is colored with the same color as a non-joint edge of `old` if both edges share no vertices and hence are parallel.

The cube coordinates are computed in the following manner. As `old` and `new` are adjacent, their cube coordinates differ by $\pm 1$ in exactly one entry of $(x,y,z)$ by Proposition 2.9. The color of the joint edge between the two cells
defines which entry needs to be changed. The decision of whether the entry differs by $+1$ or $-1$ follows from the constraint that the sum of cube coordinates, $x + y + z$, must be either 0 or 1 (see Proposition 2.9). That is,

$$\text{old has coordinate sum } 0 \quad \Rightarrow \quad \text{new has coordinate sum } 1$$
$$\text{old has coordinate sum } 1 \quad \Rightarrow \quad \text{new has coordinate sum } 0$$

After all cells have been visited and the iteration is finished, the cube coordinates are shifted by `radius`/2 (rounded down to integer value) for all three dimensions to ensure that all coordinates are positive.

## 3.3 Step 3: Preparing the simulation data

The simulation data needs to be moved from the triangular grid to the cube coordinates of the cubulation. This is achieved by the function `prepare_field` for single-level data, and by the function `prepare_field_lev` for multi-level

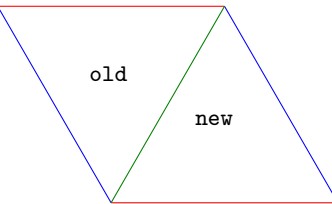

**Figure 7.** Illustration of how to color the edges of a newly visited cell (right) based on the edge colors of an old cell (left).

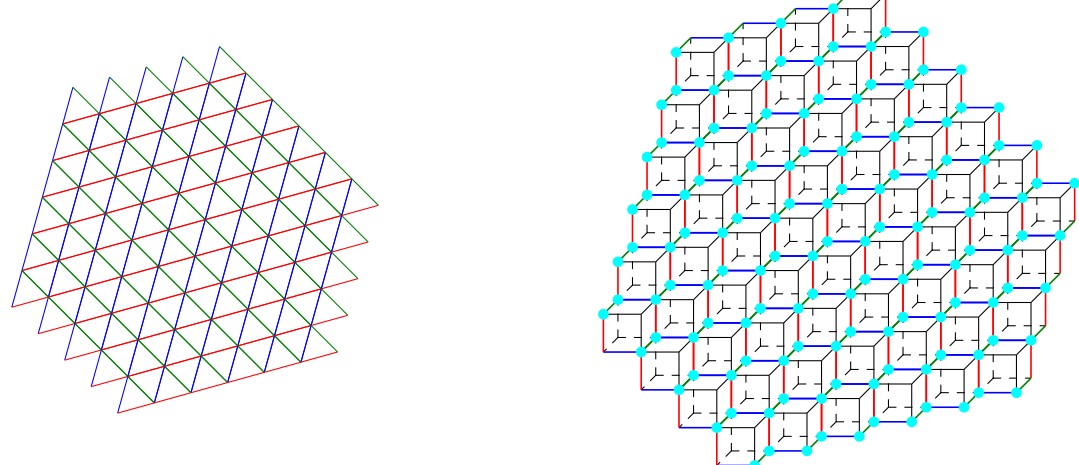

**Figure 8.** Illustration of `compute_cubulation`. The triangular grid is shown in the left panel, the corresponding cubical grid in the right panel. In the right panel, cyan vertices correspond to triangle centers in the left panel, and analogously for colored edges. Vertices with no colors in the right panel do not correspond to triangle centers.

data. The functions are wrappers for model-specific functions. We include such functions for the ICON model; writing analogous functions for other models and their data format should be straightforward.

`prepare_field` and `prepare_field_lev` require as input the cubulation computed in Step 2 and a threshold value. The latter is used to convert the input data to values of either 0 and or 1, depending on whether the input data is smaller or equal-or-larger than the threshold. The functions return the thresholded input data on the triangular grid as well as on the cubical grid, where the data on the cubical grid has dimensions $(\texttt{radius}+1)^3$ for single-level data and $\texttt{nlev}\times(\texttt{radius}+1)^3$ for multi-level data, with $\texttt{nlev}$ being the number of levels. In case of multi-level data, the 330 first entry corresponds to the model level, and the following entry describes the horizontal position. For the triangular grid, the horizontal position is given by the cell index $i$; for the cubic grid it is given by the three integers $(x, y, z)$.

   `prepare_field_lev` moves the entire multi-level data to the cubical grid, which for larger grids can result in a large memory burden (see Sect. 4). The memory burden can be easily reduced by working separately on each level by means of `prepare_field` and using the function `compute_levelmerging` to connect components in the vertical. An 335 example for how this is achieved is provided in the juypter notebook `example-3d.ipynb`

### 3.4 Step 4: Computing connected components

Once the cubulation is known and the simulation data is prepared, computing the connected components is done by the functions `compute_connected_components_2d` and `compute_connected_components_3d`, respectively. The functions require as input the cubulation (Step 2) and the prepared simulation data on the cube grid (Step 3).

Two types of connectivity in the horizontal direction can be chosen: vertex or edge connectivity. For edge connectivity, cells in the horizontal belong to the same component if they share a triangle edge. For vertex connectivity, they also belong to the same component if they share only a triangle vertex. Vertex connectivity thus results in larger but fewer connected components. The default choice is vertex connectivity. Examples illustrating edge- and vertex connectivity are provided in Figures 10 and 9.

The functions `compute_connected_components_2d` and `compute_connected_components_3d` use the external library cc3d (Silversmith, 2021) to identify connected components on a single model level. cc3d can identify connected components in three dimensions for 26 and 6-connectivity, and uses a 3-d variant of the two pass method by Rosenfeld and Pfaltz (1966) augmented with union-find and a decision tree based on Wu et al. (2005). For multi-level data the external library NetworkX (Hagberg et al., 2008) is used to merge connected components in the vertical by

constructing a corresponding graph (see Sect. 2.4) and applying breadth-first search. The final result of both functions is a list of connected components. For 2-d data a connected component is given by a list of triangular cell indices. For 3-d data, it is given by a list of tuples, with each tuple consisting of the model level and cell index.

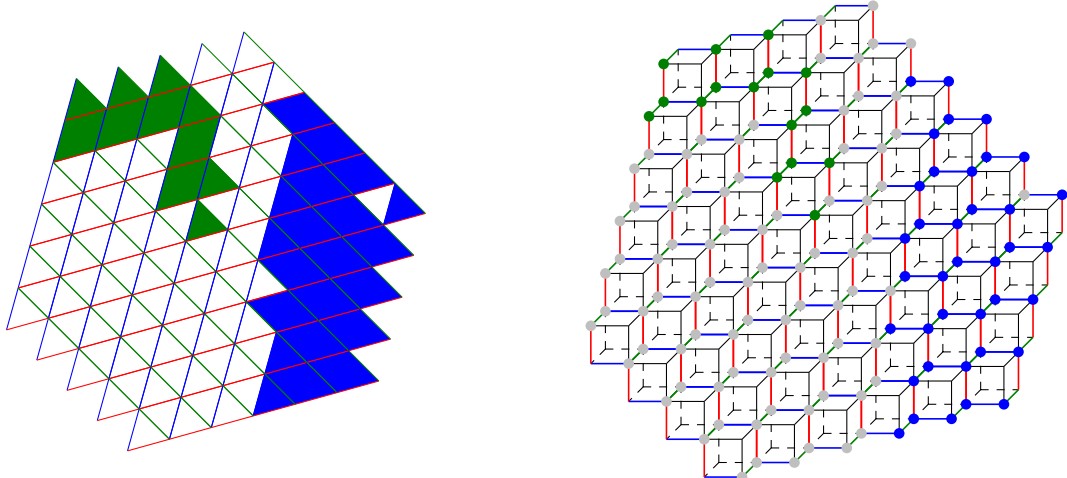

**Figure 9.** Illustration of the result of connected component labeling for 2-d data and vertex connectivity. In the left plot, connected components are formed by cells with the same face color. The colors of the cell edges that form the hyperplanes of the cubulation are also shown. The right plot illustrates the same data on the cubulation. Triangles on the left correspond to vertices on the right. A set of triangles on the triangular grid is vertex connected if in the set of vertices on the cubical grid any two vertices in the set can be connected by a sequence of vertices where subsequent ones are in a common 2-dimensional face (i.e. square) of a cube.

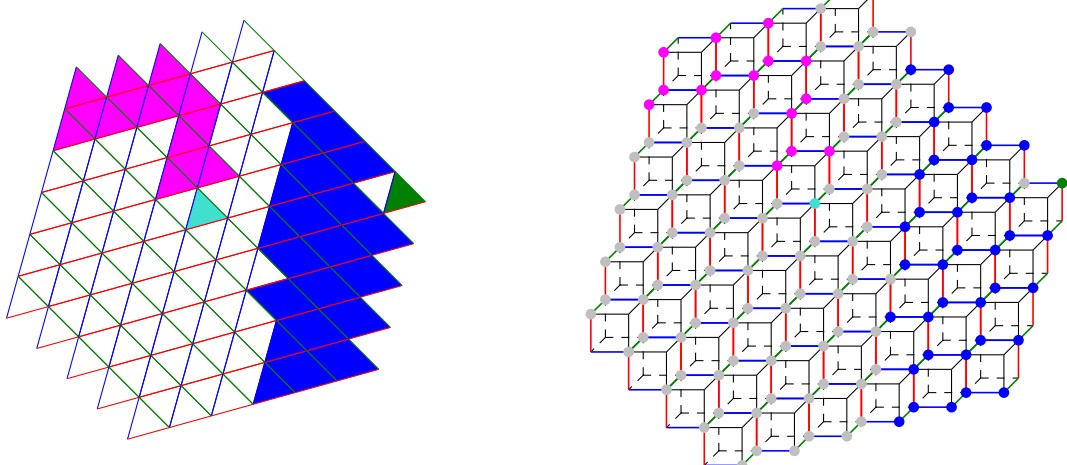

**Figure 10.** Same as in Figure 9 but for edge connectivity. A set of triangles on the triangular grid is edge connected if in the set of vertices on the cubical grid any two vertices in the set can be connected by a sequence of vertices such that two subsequent ones are connected by an edge of a cube.

## 3.5 Application examples

As an illustration of TriCCo's abilities, we analyze model output from the ICON atmosphere model in limited-area
setup over the North Atlantic. The output is from a simulation that applies a triangular grid with 7,920 cells and 75 model levels and that is part of a larger set of simulations presented in Senf et al. (2020) and Stevens et al. (2020) from a scientific perspective. In our work here, the simulation output solely serves as technical input to test and illustrate the functionality and performance of the TriCCo routine. The grid has a nominal horizontal resolution of 80 km. Its characteristics are listed in Tab. 1. We use total cloud cover for demonstrating the use for data on a single
model level, and vertically-resolved cloud fraction on 75 levels as an example for multi-level data.

The simulation domain is illustrated in Fig. 1 and extends from 78 W to 40 E, and 23 N to 80 N, covering the North Atlantic, the Mediterranean, the larger part of Europe and parts of Northern Africa. To find the start cell we search for the cell with the smallest distance to the grid centre at 19 W and 51.5 N, where we measure the distance in terms of the great circle distance on the sphere using the haversine formulae. We then find the radius by using the lower
bound and the diagnostic output of `compute_cubulation` as described in Sect. 3.2. The start cell and radius are given in Tab. 1.

Fig. 11 shows the result of connected component labeling for total cloud cover, which in ICON ranges between 0% (cloud-free) and 100% (completely cloud covered). Panel a shows total cloud cover from a single time step of the simulation. Centered at roughly 20 W and 50 N, there is a commashaped cloud band that is associated with a
370 warm-conveyor belt of a North Atlantic extratropical cyclone. We threshold the data at 85%, as shown in panel b, and identify connected components for vertex (panel c) and edge connectivity (panel d). For vertex connectivity, the

commashaped cloud band is connected to a cloud structure west of it. Using edge connectivity instead, the cloud band can be isolated. Overall, vertex connectivity leads to 31 connected components, compared to 74 components when the stricter criterion of edge connectivity is used.

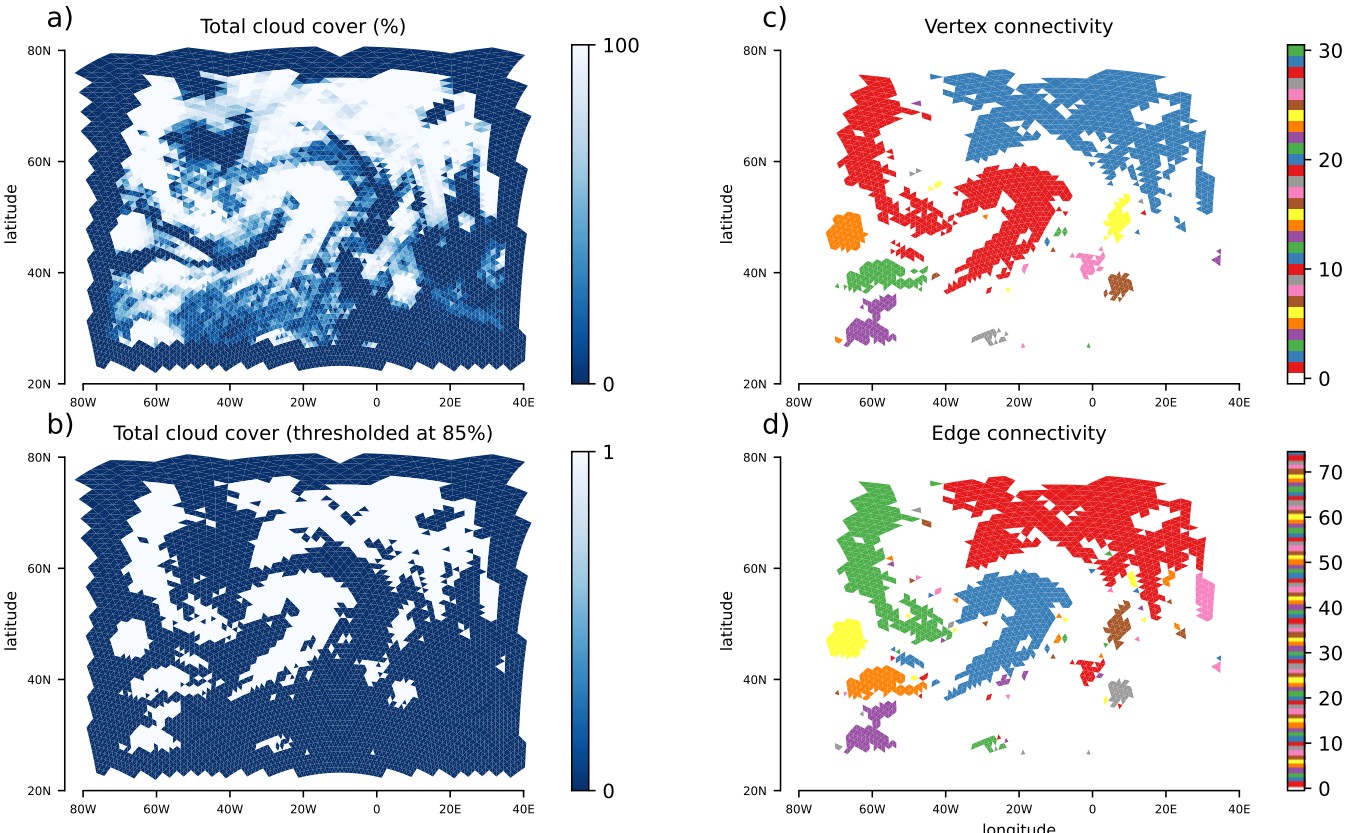

**Figure 11.** Application for total cloud cover from an ICON simulation with a limited-area grid over the North Atlantic. The triangular nature of the grid is visible. (a) Total cloud cover in per cent, with 0 corresponding to cloud-free and 100 to completely overcast conditions. (b) Total cloud cover thresholded at 85%, with values above 85% set to 1 and values below set to 0. (c) Connected components for vertex connectivity, with components being plotted in different colors. (d) Connected components for edge connectivity. In c and d, the connected components are ordered according to their size, starting with the largest component.

We also present results for connected component labeling for vertically-dependent cloud fraction from the same time step, where we again apply a threshold of 85%. For vertex connectivity we identify 235 components, whereas edge connectivity results in 381 components. Fig. 12 shows the component that corresponds to the commashaped cloud band centered at 20 W and 50 N. Note that for displaying purposes, the horizontal grid is rotated and latitude decreases from left to right. The vertical structure of the cloud band is clearly visible, as is the fact that vertex

connectivity associates more cells to the connected component than edge connectivity. This can be seen, for example, near the surface around 10 W and 40 N.

The Python code of the examples presented here is included in TriCCo as jupyter notebooks `examples-2d.ipynb` and `examples-3d.ipynb`.

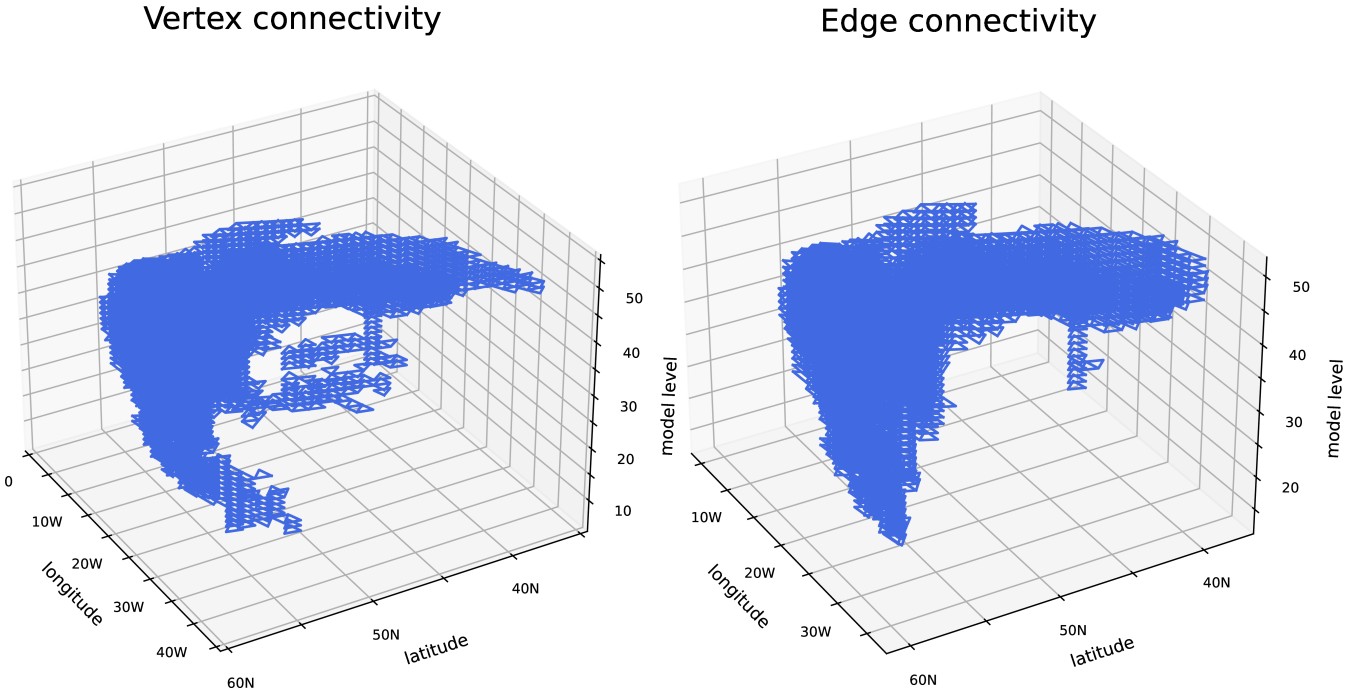

**Figure 12.** Application to vertically-varying cloud cover from the ICON simulation also used in Fig. 11. The plot shows the connected component corresponding to the commashaped cloud band near 20 W and 50 N for vertex connectivity (left) and edge connectivity (right). The model levels are counted upward from the Earth surface so that level 10 is near the surface and level 55 in the upper troposphere.

## 4   Benchmarks and computational challenges

The ICON simulation analyzed in Sect. 3.5 is part of a larger set of simulations that includes triangular grids with much finer resolution. In this section, we use this larger set to characterize the computational aspects of TriCCo, and to identify limitations of the current implementation. The limitations result, for example, from the current serial implementation that restricts use to a single core. Ultimately, they reflect that as climate scientists and pure mathematicians our expertise in software development and computational science is finite.

We use simulations with horizontal resolutions ranging from 80 km to 10 km. Their grid specifics are included in Tab. 1. Because we are interested in the computational performance, what matters here is not the grid resolution

| Horizontal resolution in km | 80 | 40 | 20 | 10 |
|---|---|---|---|---|
| *Triangular grid* | | | | |
| Cells | 7,920 | 31,728 | 127,052 | 508,988 |
| Vertices | 4,089 | 16,121 | 64,042 | 255,528 |
| Edges | 12,008 | 47,848 | 191,093 | 764,515 |
| *Cubulation* | | | | |
| Start cell | 5,570 | 18,494 | 69,220 | 264,617 |
| Search radius | 104 | 210 | 423 | 851 |
| Size of cubulation | $105^3$ | $211^3$ | $400^3$ | $802^3$ |

**Table 1.** Size of the triangular grids used for benchmarking, as well as characteristics of the associated cubulations.

itself but the number of grid cells. The latter increases by roughly a factor of four for each grid refinement. The start cells and search radii depend on the grid and we find them following the approach outlined in Sect. 3.5. The benchmarks are run on a single core of a dedicated compute node of the Levante HLRE-4 supercomputer of Deutsches
Klimarechenzentrum in Hamburg, Germany, which became operational in spring 2022. A Levante compute node is equipped with 2 AMD 7763 EPYC CPUs (64 cores, base frequency of 2.45 GHz) and 256 GB main memory.[2]

We measure the time needed for Steps 2, 3 and 4 described in Sect. 3. The computational cost for Step 1 is virtually zero and not considered. As in Sect. 3.5 we use total cloud cover for single-level data, and cloud fraction on 75 model
levels for multi-level data.

The time required to compute the cubulation (Step 2) increases strongly with the number of grid cells (Tab. 2). For the coarsest grid with 80 km resolution and 7,920 cells, the cubulation is computed within a few seconds; for the 10 km grid with 508,988 cells this step takes 4 hours. We find this result encouraging as it shows that even our rather naive implementation can handle grids that contain up to 500,000 cells. To put that number into perspective,
global simulations with ICON in climate mode are run using grids with 20,480 cells (R2B4 resolution; Giorgetta et al., 2018), and global simulations with ICON in weather-prediction mode for research purposes typically use grids with 327,680 cells (R2B6 resolution; Selz, 2019; Baumgart et al., 2019). Such grids are accessible already with our current implementation, despite a clear need for improvement if one aims to handle larger grids, including those used in global storm-resolving models (Satoh et al., 2019). However, it is also important to keep in mind that the
cubulation needs to be computed only once for a given grid. Hence, the time for computing the cubulation itself is not a limiting factor for use cases with many time steps or simulations done on the same grid.

---

[2]https://docs.dkrz.de/doc/levante/configuration.html

| Horizontal resolution in km | 80 | 40 | 20 | 10 |
|---|---|---|---|---|
| Step 2: Compute cubulation | 12 s | 82 s | 15 m | 4 h |
| *Single-level data* | | | | |
| Step 3: Read and prepare data | 0.05 s | 0.1 s | 0.3 s | 1 s |
| Step 4: Vertex connectivity | 0.03 s | 0.1 s | 0.6 s | 3 s |
| Step 4: Edge connectivity | 0.03 s | 0.1 s | 0.6 s | 3 s |
| *Multi-level data (75 levels)* | | | | |
| Step 3: Read and prepare data | 0.5 s | 2 s | 9 s | 38 s |
| Step 4: Vertex connectivity | 3 s | 14 s | 70 s | 370 s |
| Step 4: Edge connectivity | 4 s | 24 s | 130 s | 630 s |

**Table 2.** Time required for different aspects of TriCCo's Python implementation for different sizes of the triangular grid. The times are obtained from benchmarks on the DKRZ Levante supercomputing system.

Tab. 2 also includes the times required for reading and preparing the simulation data (Step 3), and for computing connected components (Step 4). In real-world applications of TriCCo, both steps are done together. Here, their times are separated to help identify performance bottlenecks. The times are obtained from 10 repetitions of analyzing 48 output time steps and are given as the average time required for a single time step. An exception is the 10 km grid for multi-level data, for which the time is obtained from a single analysis of 10 time steps due to the rather large computational expense and the 8-hour wall-clock limit for a batch job on Levante's compute partition. All times are given per single time step.

The time for computing the connected components dominates the time for reading and preparing the data. For single-level data, computing the connected components requires only a few seconds even for the 10 km grid and in fact takes (much) less than 1 s for the smaller-sized grids. This shows that the current implementation is feasible for single-level data. For multi-level data the picture is mixed. The time stays within a few seconds for the 80 km grid but increases strongly as the grid size is increased. For the 10 km grids analyzing a single time step takes up to 10 minutes. In practice, this may limit the application of TriCCo to grids of this size and larger.

Besides speed, another matter of concern is the amount of required main memory. In the current implementation, the entire data on the cubic grid needs to be hold in memory. The size of this data increases with the size of the cubulation, which is included in the last line of Tab. 1. For example, the cubulation of the 10 km grid consists of $802^3 = 516 \cdot 10^6$ cells, meaning that the cubulated version of single model-level data requires approximately 500 MByte of memory if one assumes the data is stored as 1-Byte integers. For multi-level data on, e.g., 75 levels, the requirement increases to 36 GByte, although the memory requirement can be reduced to essentially that for single-level data by

treating levels separately as described in Sect. 3.4. Nevertheless, TriCCo's thirst for memory can become immense. While the memory requirement can be satisfied on high-performance computers (for example, a DKRZ Levante compute node has 256GB of memory), this might pose a problem for the general applicability of TriCCo and for

grids larger than those considered here.

The need to reduce the amount of required memory is also evident from a consideration of information density. On the triangular grid, each cell contains information and the information density is maximum. When the data is moved onto the cubic grid, the vast majority of cells in fact do not correspond to a cell on the triangular grid and contain no information. I.e., the information density is very low. A striking example is the 10 km grid, for which out of the

516 million cells of the cubic grid only 508,988 correspond to a cell on the triangular grid, that is less than 0.1%. Put differently, the data moved to the cubic grid data is a very sparse matrix whose entries for the overwhelming part are trivial 0s.

## 5    Comparison to alternative methods

To put TriCCo's performance into context, we test two alternative methods. The first is breadth-first search that we

have implemented ourselves in Python. The second makes use of the NetworkX library, which provides functions to build a graph and search for its connected components via breadth-first search. Breadth-first search is a well established simple algorithm for searching a graph and is guaranteed to find all connected components (Cormen et al., 2009). The choice of the NetworkX library is pragmatic since we already use it in TriCCo for merging connected components in the vertical (see Sect. 3.4). Other graph libraries might be faster than NetworkX (Staudt et al., 2014),

yet our aim here is not to provide an exhaustive comparison of TriCCo to other methods and implementations, of which there are many (He et al., 2017). Rather, by choosing NetworkX we emphasize the perspective of a TriCCo user that we have in mind: a geophysical modeler and data analyst with good experience in Python but little background in computer science and computer vision (e.g., the first author of this paper).

The two alternative methods are included in TriCCo's repository; their use is documented via the juypter

notebooks `alternative_own-bfs_2d.ipynb` and `alternative_networkx_2d.ipynb`. We only consider single-level data. Performance differences for single-level data carry over to multi-level data in a straightforward manner because multi-level data is treated by looping over single-level data followed by level merging. As in Sect. 4 the two methods are benchmarked on the DKRZ Levante system, with the times reported in Tab. 3. For both methods, the time to read and prepare (i.e., threshold) the data is not listed as it remains below 0.1 s for all grids. The read and prepare

step is faster compared to TriCCo as the data remains on the triangular grid and does not need to be moved to the cubic grid.

Regarding our own breadth-first search, the timing is clearly disadvantageous compared to TriCCo (Tab. 3). E.g., for the 20 km grid a single time step takes more than 1 minute for vertex connectivity, which is more than 100 times

| Horizontal resolution in km | 80 | 40 | 20 | 10 |
|---|---|---|---|---|
| *Own breadth-first search* | | | | |
| Vertex connectivity | 2 s | 12 s | 100 s | - |
| Edge connectivity | 0.5 s | 3 s | 20 s | - |
| *Networkx* | | | | |
| Prepare full graph for vertex connectivity | 3 s | 11 s | 50 s | 250 s |
| Prepare full graph for edge connectivity | 2 s | 8 s | 50 s | 430 s |
| Vertex connectivity | 0.8 s | 3 s | 13 s | 55 s |
| Edge connectivity | 0.2 s | 0.6 s | 3 s | 10 s |

**Table 3.** Time required for different aspects of two alternative methods for different sizes of the triangular grid. The times are obtained from benchmarks on the DKRZ Levante supercomputing system.

slower than TriCCo. The approach, while simple and short in terms of programming, quickly becomes impractical as
the grid becomes larger. As a result, we have not tested it for the grid with 10 km resolution.

The NetworkX-based method consists of two steps. First, the graph with all grid cells as nodes and all links between adjacent cells is constructed[3]. We refer to this as the "full" graph. The full graph needs to be constructed only once for vertex and edge connectivity, respectively. One can think of the construction of the full graph as the analogue of TriCCo's cubulation, as both need to be computed only once for a given grid. This step is cleary faster in
NetworkX compared to TriCCo (Tab. 3).

In a second step, for analyzing a single time step, the nodes that correspond to cells with no valid data after thresholding (i.e., cells with values of 0) are removed from the full graph, and the resulting time-step specific graph is used for the identification of connected components via NetworkX. The time for finding connected components increases roughly linearly as a function of the number of links, and vertex connectivity takes longer than edge
connectivity for the same grid size (Tab. 3). Both findings are expected for breadth-first search (Staudt et al., 2014). However, the computation via NetworkX is a factor of 3 (for edge connectivity) and 20 (for vertex connectivity) slower compared to TriCCo. Thus, for use cases with many time steps TriCCo can provide a substantial advantage compared to NetworkX.

We note that our comparison focuses on run time. An advantage of the NetworkX method (and analogous
graph-based implementations) is that it requires much less memory than TriCCo. Depending on the size of the data and the available computer this might be relevant when applying TriCCo.

---

[3]In graph theory, the links are often called edges. Here we use the term link to avoid confusion with edge connectivity.

# 6 Discussion and conclusions

In this work we have developed a new method for identifying connected components for data on triangular grids. The key principle of our method is to map the triangular grid to a structured cubic grid. For the cubic grid, neighbor relationships are encoded directly in the cell indices. As a result, the connected components can be identified by previously developed algorithms for connected component labeling on cubic grids. Furthermore, we have provided a Python implementation of the method named TriCCo, illustrated its use, and characterized its computational performance.

The most expensive step of our method is the computation of the mapping between the triangular grid and the cubic grid, i.e., the cubulation. In TriCCo's current implementation, this step can be time consuming for large grids. We suspect this step could be accelerated relatively easily by someone with more expertise in scientific programming, for example by avoiding type changes or moving the implementation from pure Python to C++. However, it is also important to remember that the cubulation only needs to be done once for a given grid, meaning that its relative cost decreases as more time steps or simulations with the same grid are analyzed.

A key feature of our method is that it allows us to use previously developed software libraries that identify connected components on cubic grids. Cubic grids have found to be of advantage for problems in, e.g., robotics (Ardila et al., 2014), and so there might be reason to believe that these advantages might carry over to connected component labeling on cubical grids compared to approaches that directly work on the graph constructed by the triangle cells and their adjacency information. We have shown that our specific implementation that uses the external library cc3d (Silversmith, 2021) for connected labeling on the cubic grid performs better than a graph-based approach with the NetworkX library (Hagberg et al., 2008). This finding, however, is difficult to generalize and ultimately depends on the computational performances of the libraries applied for the cubic grid versus the graph approach. NetworkX is implemented in pure Python; for example it might well be that the NetworKit library (Staudt et al., 2014), which uses C++, includes OpenMP parallelization and provides a Python API, would beat TriCCo. At the same time, TriCCo could be accelerated by replacing cc3d with a different library that for example could exploit the fact that most of the cubes are trivial zeros, thereby accelerating the raster scan of Rosenfeld and Pfaltz (1966). The upshot of these considerations is that we cannot and do not intend to claim that TriCCo beats existing algorithms, but that TriCCo works with reasonable speed (apart from the cubulation step) and provides a new approach to connected component labeling on triangular grids.

Our application examples of TriCCo use a limited-area triangular grid from the ICON model. In its global version, ICON uses a triangular grid based on the icosahedron projected onto the sphere (Wan et al., 2013; Zängl et al., 2015). It would be straightforward to extend TriCCo to such a global grid by mapping the icosahedron to the triangulated plane using a net, which in essence means to cut open and unfold the icosahedron along its edges. Such a net can then be embedded into a regular triangle grid with six triangles around every vertex. To compute connected components on such a net one can then use TriCCo with the additional information of the sides of the net that need to be identified

to form the icosahedron. It would also be possible to extend TriCCo to an ICON grid with regional refinement as shown in Fig. 1 b of Ullrich et al. (2017). In this case, one could compute the cubulation separately for the coarse and refined parts of the grid and introduce a routine that matches coarse and refined triangles at the boundary of the refined region. However, general triangulations cannot be handled by TriCCo, as the construction of the hyperplanes used by TriCCo requires that each vertex belongs to six triangles.

TriCCo is made available open source. We welcome contributions from data and computational scientists to study if and how TriCCo can be improved. At the same time, we welcome climate and atmospheric scientists as well as more broadly colleagues from other geoscientific disciplines to use TriCCo and see if it can be of benefit to their research.

*Code availability.* The Python implementation of TriCCo is available at https://gitlab.phaidra.org/climate/tricco and can be installed from pypi. The gitlab repository contains example scripts and ICON example data that illustrate the application of TriCCo and reproduce Figs. 11 and 12. Version 1.1.0 described and used in the revised version of the paper is long-term archived at zenodo with doi:10.5281/zenodo.6667862. The previsious version 1.0.0 used in the orginal submission is archived at zenodo with doi:10.5281/zenodo.5774313.

*Author contributions.* AV and PS initiated, conceptualized and administered the project. NR and NK developed an initial Python implementation of TriCCo that was then further developed and curated by AV. PS developed the mathematical aspects with support by NR and NK, AV led the application aspects with support from NR and NK. All authors wrote, edited and reviewed the manuscript.

*Competing interests.* The authors declare that they have no conflict of interest.

*Acknowledgements.* This research was supported by a YIN Award grant from the Young Investigator Network of the Karlsruhe Institute of Technology and by seeding funding from the Centre MathSEE: Mathematics in Sciences, Engineering, and Economics of Karlsruhe Institute of Technology. AV acknowledges supported by the German Ministry of Education and Research (BMBF) and FONA: Research for Sustainable Development (www.fona.de) under Grant Agreement 01LK1509A. The TriCCo package was developed and tested on the Mistral and Levante supercomputers of the German Climate Computing Center (DKRZ) in Hamburg, Germany, using compute resources from the DKRZ project bb1018.

We are extremely thankful to the communities of developers and maintainers of the open source Python packages NumPy (Harris et al., 2020), xarray (Hoyer and Hamman, 2017), cc3d (connected-components-3d) (Silversmith, 2021), NetworkX (Hagberg et al., 2008), Matplotlib (Hunter, 2007) and Plotly, which are all used in the TriCCo package. We also thank the Phaidra service of University of Vienna for hosting the gitlab repository.

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
