# Peer review of "TriCCo v1.1.0 - a cubulation-based method for computing connected components on triangular grids"

_Geoscientific Model Development, 2021_

## Referee Comment (RC2)

Review of GMD-2021-349

**Title:**
*TriCCo v1.0.0 – a cubulation-based method for computing connected components on triangular grids*

**Authors:**
*Aiko Voigt, Petra Schwer, Noam von Rotberg and Nicole Knopf*

This is a well written manuscript that details the application of a novel algorithm for 3D coordinate labelling of triangular meshes based on embedding these meshes into the space of stacked cubes. While I enjoyed reading through the manuscript, the authors are not particularly convincing about how their algorithm is an improvement among other simpler and more efficient methods. The manuscript largely comes across as describing a mathematical curiosity (i.e., from the right angle a cube looks like a hexagon) rather than an algorithm that one might expect to adopt in practice. My concerns about proper motivation fit into three categories:

**Connectivity:** It appears that the cubulation-based method does not aid in the identification of connectivity in a given triangular grid. Indeed the algorithm described for the construction of the cubulation is based on a graph search of the dual grid. However, construction of the dual grid requires that we already know the connectivity between triangles of the grid, so that graph edges could be identified between these regions. The authors seem to engage in a bit of circular logic to justify their approach: "Because [triangular meshes] are unstructured, neighbor relations are not self-evident and identifying connected components is challenging. Our method addresses this challenge by involving the mathematical tool of cubulation." But the method itself doesn't actually address this challenge since the connectivity information needs to be known a priori to build the cubulation.

**Connected Components:** While the authors have clearly shown that the cubulation can be used to identify connected components, no significant effort has been made to compare and contrast their approach with other simpler, robust and efficient algorithms from the literature. For instance, consider the review by He et al. (2017). As mentioned in the previous paragraph, since the connectivity information must be known prior to computing the cubulation, simple O(n) graph search algorithms applied to this connectivity graph can be used for both edge connectivity and vertex connectivity as well. These simple graph search algorithms also have well-known and efficient parallel implementations (e.g., Wu et al., 2009).

He, L., Ren, X., Gao, Q., Zhao, X., Yao, B. and Chao, Y., 2017. The connected-component labeling problem: A review of state-of-the-art algorithms. Pattern Recognition, 70, pp.25-43.

Wu, K., Otoo, E. and Suzuki, K., 2009. Optimizing two-pass connected-component labeling algorithms. Pattern Analysis and Applications, 12(2), pp.117-135.

[Figure]

Figure 1: (Left) A regionally refined mesh in the ICON model providing finer resolution over Europe. (Right) A triangular voronoi tesselation and its dual from the OLAM model, here capturing a smooth transition between mesh resolutions.

**Robustness:**  While the authors emphasize early on that their study is focused on unstructured meshes, the only examples given in the text are structured triangular meshes (i.e., isometric grids consisting of equilateral triangles). Is this algorithm applicable to common refined triangular grids used in atmospheric science, such as those depicted in Figure 1?

Given that there is nothing fundamentally incorrect about the manuscript, I am recommending this paper for major revisions, but expect that in a subsequent revision that the authors address the points above and provide some comprehensive justification of how their method is an improvement to existing analysis methods.

---

## Author Response (AR1)

Geosci. Model Dev. Discuss., author comment AC1
https://doi.org/10.5194/gmd-2021-349-AC1, 2022

[Figure]

**Comment on gmd-2021-349**

Aiko Voigt et al.

Author comment on "TriCCo v1.0.0 – a cubulation-based method for computing connected components on triangular grids" by Aiko Voigt et al., Geosci. Model Dev. Discuss., https://doi.org/10.5194/gmd-2021-349-AC1, 2022

**Answer letter to the reviewers' comments**

**Aiko Voigt on behalf of all authors**

First of all, we would like to thank both reviewers for their sincere interest in our work and their very helpful remarks, which have motivated us to substantially revise the paper. We are confident that our revision answers the comments in a satisfying manner, and hope that our revised paper will be acceptable for publication in GMD. In particular, we have adapted the introduction and the conclusion (now named discussion and conclusion, section 6) to better carve out the motivation for our new method, and to characterize its strength and limitations. We have also added a new section 5 to compare TriCCo to alternative methods based on graphs. We have also made some smaller editorial changes to increase readability, and have changed figure 11 so that the connected components are now ordered by size (which we think makes the figure somewhat easier to read). When submitting the revised manuscript we intend to also submit a tracked-changes version (assuming this is possible at GMD) to help the reviewers to quickly spot our text changes. In the following, we describe our changes in response to each reviewer comment in more detail.

**Reviewer 1:**

General comment: "One major issue puzzles me. I apologize if I've just missed something obvious. The content of this review focuses on this issue. If I've missed something obvious, simply explain and that will be sufficient. If my point is substantial, then I'd like the revised manuscript to state clearly at the outset that alternative, much more efficient, methods exist to solve the problem of finding connected components of a triangulation. Again, the text itself is well written, and even if I'm right that the particular problem of finding connected components is much more efficiently done using standard graph-theoretic means, I suggest the text be published to document this cubulation procedure in case it is of use in other contexts."

Answer: Thank you, we agree that other graph based approaches exist. This is now clearly stated in the introduction, and also has motivated us to include the new section 5. In this new section, we compare TriCCo in particular to a graph-based approach using the

NetworkX library. We find that notwithstanding the cubulation step, which is expensive, the actual connected component labeling is faster in TriCCo. However, this comparison remains limited, and in the reworked discussion and conclusion section we now point out this limitation. We also would like to stress that while the cubulation step is expensive, its relative cost is slow as it only needs to be done once for a given grid; it also seems the cubulation step could be accelerated relatively easily by someone with more computer science expertise. These points are made clear in several places of the revised manuscript (e.g., section 6, abstract).

Specific comment 1: "1. According to Table 2, the most expensive part of the workflow by a large factor is the cubulation."

Answer: As written above, it is important to remember that the cubulation needs to be done only once, and so its relative cost decreases strongly as more time steps or simulations with the same grid are analyzed.

Specific comment 2: "As a smaller issue, the figures and text imply the triangulation needs to be geometrically regular so that the index scheme works. Is this a true constraint? That is, will this method not work for geometrically unstructured triangular grids?"

Answer: Yes, the reviewer is right here. Our method cannot handle general triangulations of the sphere, as the construction of the hyperplanes requires that each vertex has six triangles. We now state this limitation in the conclusion sections.

Typos: Thanks! We have corrected them, as well as others that we found during the revision.

**Reviewer 2:**

Comment: "While I enjoyed reading through the manuscript, the authors are not particularly convincing about how their algorithm is an improvement among other simpler and more efficient methods."

Answer: Yes, this was a weak point of our original manuscript. We have reworked the introduction to better describe why we have developed TriCCo and why we believe it represents a useful addition to the connected component labeling problem. The latter point is also made stronger in the discussion and conclusion section. Furthermore we have added a new section 5 to compare TriCCo to graph-based methods using breadth-first search. In this section we find that TriCCo compares well, but we also point out the limitations of this comparison in the discussion and conclusion section (section 6).

Comment on "Connectivity": "The authors seem to engage in a bit of circular logic to justify their approach: "Because [triangular meshes] are unstructured, neighbor relations are not self-evident and identifying connected components is challenging. Our method addresses this challenge by involving the mathematical tool of cubulation." But the method itself doesn't actually address this challenge since the connectivity information needs to be known a priori to build the cubulation."

Answer: We believe this was a misunderstanding due to our unclear writing. Of course the adjacency information must be available also on the triangular grid. The advantage of the cubic grid is that the adjacency information is directly encoded in the cube indices. This is now written clearly in the revised introduction section and the revised discussion and conclusion section.

Comment on "Connected Components": "While the authors have clearly shown that the

cubulation can be used to identify connected components, no significant effort has been made to compare and contrast their approach with other simpler, robust and efficient algorithms from the literature."

Answer: Thanks, we agree that this was a shortcoming of our paper. In the revision we have added a new section 5 to attempt such a comparison, even though we admit in the discussion in section 6 that our comparison might be biased by the choice of external libraries (cc3d versus NetworkX). As a consequence, we stress that in section 6 that cannot and do not intend to claim that TriCCo beats existing algorithms, but rather that our message is that that TriCCo works with reasonable speed (apart from the cubulation step) and provides a new approach to connected component labeling on triangular grids." We have included He et al. (2017) as a reference.

Comment on "Robustness": "While the authors emphasize early on that their study is focused on unstructured meshes, the only examples given in the text are structured triangular meshes (i.e., isometric grids consisting of equilateral triangles). Is this algorithm applicable to common refined triangular grids used in atmospheric science, such as those depicted in Figure 1?"

Answer: Thank you, this is an important point. Our method can be quite easily extended to the global ICON grid and to regional refinements, but general triangulations cannot be handled. We now devote a new paragraph in the discussion and conclusion section (section 6) to clarify this point.

**Changes independent of the reviewer comments:**

Fig. 11: We have now sorted the connected components according to their size so that the largest connected component corresponds to the color with value 1.

Benchmarking: We have redone the benchmarks on the new DKRZ Levante system. The previously used Mistral system was decommissioned in May 2022. The numbers do not change substantially.

In section 3.3., L318-322 were an old spurious textblock that we now removed. Also, the caption of Fig. 8 was changed as "make_cubical_coordinates" was an old outdated function name and actually needs to read "compute_cubulation".

The TriCCo version for the revised paper will be 1.1.0. The zenodo archive will be updated accordingly, as well as the pypi package.